# Evaluation of antibiotic treatment initiation and duration practices in primary care. Cross-sectional study in two French multi-professional health centers

**Simon Ferme[1]☯, Emmanuel Piednoir[1,2]☯, Marianne Delestre[1]‡, Elise Fiaux[1]‡, Johann Masik[1]‡, Renaud Verdon[2,3]☯, Pascal Thibon[1]☯ \***

1 Centre Régional en Antibiothérapie Normandie, CRAtb «Normantibio», Centre Hospitalo-Universitaire, Caen, Normandie, France, 2 UNICAEN, UNIROUEN, Inserm UMR 1311 DYNAMICURE, Normandie Univ, Caen, France, 3 Service des Maladies Infectieuses et Tropicales, Centre Hospitalo-Universitaire, Caen, Normandie, France

☯ These authors contributed equally to this work.
‡ MD, EF and JM also contributed equally to this work.
* thibon-p@chu-caen.fr

**Data Availability Statement:** All relevant data are within the manuscript and its Supporting information files.

## Abstract

Antibiotic resistance poses a significant human and economic burden. In France, which ranks among the highest consumers of antibiotics in Europe, 93% of prescriptions are issued in primary care, primarily for respiratory tract infections. It is crucial to limit both the indications and the duration of antibiotic prescriptions, with recently updated recommendations in France aimed at achieving this goal. Our main objective was to evaluate whether general practitioners' antibiotic initiation and prescription durations for respiratory infections align with these recommendations. In this prospective cross-sectional study conducted over six weeks in two multi-professional health centers, all consultations for respiratory infections (in both adults and children) documented in patients' medical records were reviewed. Overall, 46.8% (N = 334/714) of consultations resulted in an antibiotic prescription (15.8% for nasopharyngitis and 83.5% for acute cough and bronchitis). Compliance with recommended antibiotic durations was observed in 66.7% (N = 476/714) [95% CI: 63.1%-70.0%] of consultations, with adherence rates exceeding 80% for nasopharyngitis and pharyngitis but falling below 20% for community-acquired pneumonia and acute cough and bronchitis. In total, 1,194 excess days of antibiotic therapy were identified, with an average excess of 1.7 days per prescription [95% CI: 1.4–1.9]. There remains significant room for improvement in both reducing the initiation of antibiotic treatments and shortening their prescribed durations. Managing acute coughs and bronchitis continues to be one of the key challenges in primary care. For nasopharyngitis, the high frequency of this condition translates into potentially large prescribing volumes on a collective scale. Efforts to promote the new paradigm of "shorter is better" for antibiotic prescription durations need to be intensified.

**Funding:** The author(s) received no specific funding for this work.

**Competing interests:** The authors have declared that no competing interests exist.

## Introduction

Antibiotic resistance (AMR) is recognized by the World Health Organization as one of the greatest global threats [1]. Emerging resistance mechanisms are impairing the treatment of common infections, resulting in complications, hospitalizations, disabilities, and even death. The human and economic impacts are substantial: in 2021 alone, 1.14 million deaths world-wide were directly attributed to antibiotic resistance, and projections indicate that nearly 2 million deaths could occur by 2050 [2].

Numerous studies have established a clear link between the volume of antibiotic consumption and the emergence of resistance [3–5]. Additionally, antibiotic use is associated with adverse events in 20% of patients [6]. The U.S. Food and Drug Administration has recently issued warnings regarding fluoroquinolones due to potential risks, including tendinopathy, hypoglycemia, mental health side effects, and an increased risk of aneurysm and aortic dissection [7].

In France, policies targeting both the general population and healthcare professionals to reduce the number of initiated antibiotic treatments have been implemented successively, but with limited or temporary effects. On a European scale, France remains one of the countries with the highest antibiotic consumption, ranking 25th out of 29 in 2021 [8]. Although primary care consumption decreased in 2020 due to the unique context of the COVID-19 pandemic, an increase was observed in 2021 and 2022. Combating antibiotic overuse remains a significant challenge in France, where 93% of prescriptions are issued in primary care settings, 71% of which are prescribed by general practitioners (GPs) [9]. The majority of community prescriptions are related to respiratory tract infections, despite their predominantly viral etiology (>70%) [10].

The indications for antibiotics must be restricted, and the duration of prescriptions should also be carefully managed. Recommendations regarding the duration of antibiotic therapy have recently been updated by the French Society of Infectious Pathology (SPILF) and the Pediatric Infectious Pathology Group (GPIP) [11].

Unfortunately, communication efforts regarding the latest recommendations on antibiotic durations have been limited due to the COVID-19 health crisis, and their adoption by general practitioners (GPs) has not yet been evaluated, to the best of our knowledge. The primary aim of this study was to assess the appropriateness of antibiotic prescribing by GPs, focusing on both initiation and duration, in accordance with the recommendations for upper and lower respiratory tract infections.

## Materials and methods

This prospective multicenter cross-sectional study was conducted between March 1 and April 9, 2022, over a period of six weeks, in two multiprofessional health centers (MHCs) comprising 12 general practitioners (GPs). The two MHCs were located in a commune with 4,000 inhabitants and a commune with 15,000 inhabitants, both situated in the Normandy region (population of 3,317,000 in 2023).

Patients who consulted one of the two MHCs during the study period were included if their reason for consultation corresponded to an upper or lower respiratory infection (Table 1), whether they were adults or children.

If a patient visited multiple times for the same infection (due to worsening symptoms or follow-up requested by the physician), only the first consultation was considered. Excluded from the study were infections requiring specialized consultations or hospitalization from the outset, SARS-CoV-2 (COVID-19) infections, and consultations where the duration of the prescription could not be determined due to it not being recorded in the medical record. At the end of the study period, data on patient characteristics, indications, and prescribed antibiotics were extracted through a query in the GPs' software.

**Table 1. Upper and lower respiratory infections in adults and children: Antibiotic treatment durations recommended by the SPILF (Société de pathologie infectieuse de langue française) and the Pediatric Infectious Pathology Group (GPIP), 2021.**

| Indication | SPILF/GPIP recommendations, 2021 |
|---|---|
| Acute exacerbation of COPD | 5 days |
| Community acquired pneumonia | 5 days (extend to 7 days if no clinical improvement is observed by day 3). |
| Acute cough and bronchitis | No antibiotic treatment (0 day) |
| Nasopharyngitis | No antibiotic treatment (0 day) |
| Group A *streptococcus* pharyngitis | 6 days (amoxicillin) |
| | For patients with a non-severe allergy to penicillin: 5 days (cefpodoxime proxetil) or 4 days (cefuroxime axetil). |
| | For patients with a severe allergy to β-lactams: 5 days (josamycin or clarithromycin), or 3 days (azithromycin). |
| Pharyngitis, RADT negative | No antibiotic treatment (0 day) |
| Acute otitis media | 5 days (>2 years and adults) |
| | 10 days (≤2 years) |
| Congestive otitis or serous otitis | No antibiotic treatment (0 day) |
| Acute sinusitis* | 7 days (amoxicillin) |
| | 5 days (anti-pneumococcal fluoroquinolones or C2G/C3G) |
| | 4 days (pristinamycin) |
| | 10 days (children) |

COPD = chronic obstructive pulmonary disease

*The choice of antibiotic depends on the site of infection (maxillary, frontal, ethmoidal, or sphenoidal) or specific circumstances, such as an associated dental infection or forms at risk of complications.

The study was approved by the local ethics committee of Caen University Hospital, which waived the requirement for informed consent. Subjects were informed about the study through written notices placed in the waiting rooms of the MHCs and were invited to inform their physician during the consultation if they chose to opt out, so that a dedicated register could be maintained. The study was conducted within the context of routine care, with no additional examinations beyond standard practice, and the diagnosis used was the one made by the general practitioner. Written consent from all GPs at the two participating MHCs was obtained following a presentation of the project.

For each consultation, the prescribed duration of antibiotic therapy (recorded as 0 days if no antibiotics were prescribed) was compared with the recommended durations [12], as outlined in Table 1. In cases where assessing the appropriate duration of antibiotic therapy proved challenging, the opinions of two infectious disease specialists who independently reviewed the case (EF, EP) were sought. For Group A *streptococcus* pharyngitis diagnosed without the use of a rapid antigen detection test (RADT), the following rule was applied: if an antibiotic was prescribed, the indication was deemed valid, and the duration was assessed according to the guidelines for pharyngitis with a positive RADT.

## Statistical analysis

Assuming a proportion of antibiotic prescriptions of 50% for upper and lower respiratory infections and an 80% adequacy rate for treatment durations when prescribed, the minimum required number of observations was 492, with a 5% alpha risk and a precision of ±5%.

Only an overall analysis was conducted; evaluations at the individual or practice level were not included for confidentiality reasons. Categorical variables were described using counts and percentages, while numerical variables were summarized by their median and interquartile range. The exact 95% confidence interval was calculated for the rate of treatment duration adequacy both overall and within each diagnostic category, as well as for the proportions of antibiotic therapy exceeding specific thresholds based on the type of infection. Excess antibiotic therapy duration was determined by the difference between the prescribed duration and the recommended duration. Statistical analyses were performed using R software, version 4.0.

## Results

During the study period, a total of 4,843 visits were recorded at the two participating MHCs, of which 784 visits (16.2%) were for upper or lower respiratory tract infections, involving 704 patients (none of whom refused to participate). A total of 70 consultations (8.9%) were excluded from the evaluation of antibiotic therapy duration for the following reasons: repeat visits for the same issue (N = 66), immediate specialist advice or hospitalization was proposed (N = 3), or the duration of therapy could not be determined (N = 1). Thus, 714 consultations were evaluated, with five cases requiring independent review by two infectious disease specialists. The number of consultations by type of infection is presented in Table 2.

Nearly half of the consultations resulted in an antibiotic prescription, with 15.8% for nasopharyngitis and 83.5% for acute cough and bronchitis (Table 2 and Fig 1). The median duration of antibiotic therapy was often longer than recommended. Overall, 66.7% (476/714) [95% CI: 63.1%-70.0%] of the evaluated consultations adhered to recommended antibiotic durations, though significant disparities were observed based on the type of infection (Table 2 and Fig 1). Compliance exceeded 80% for nasopharyngitis (Fig 1C) and pharyngitis with both negative and positive RADTs, while it fell below 20% for community-acquired pneumonia (Fig 1A) and acute cough and bronchitis (Fig 1C). Among the 75 cases of Group A *streptococcus* pharyngitis, the RADT was used 22 times (29.3%); no antibiotics were prescribed when the RADT was negative, whereas antibiotics were prescribed in 94.3% of cases (50/53) when the

**Table 2. Upper and lower respiratory infections in adults and children: Antibiotic treatment durations by type of infection.**

| Condition | N | ATB prescriptions N (%) | ATB duration (days) | Duration compliant with recommendations* | | Excess days N (%) |
|---|---|---|---|---|---|---|
| | | | Median (IQR) | N | [95%CI] | |
| Acute exacerbation of COPD | 7 | 7 (100) | 8 (7–8) | 0 | 0 [0–41.0] | 19 (1.6) |
| Community acquired pneumonia | 12 | 12 (100) | 8 (7–9) | 2 | 16.7 [2.1–48.4] | 37 (3.1) |
| Acute cough and bronchitis | 109 | 91 (83.5) | 7 (6–7) | 19 | 17.4 [10.8–25.9] | 609 (51.0) |
| Rhinopharyngitis | 400 | 63 (15.8) | 0 (0–0) | 337 | 84.2 [80.0–87.5] | 413 (34.6) |
| Group A *streptococcus* pharyngitis | 7 | 7 (100) | 6 (5–6) | 5 | 71.4 [29.0–96.3] | 1 (0.1) |
| Pharyngitis, RADT negative | 15 | 0 (0) | - | 15 | 100 [79.2–100] | 0 (0) |
| Pharyngitis, RADT not performed | 53 | 50 (94.3) | 6 (6–6) | 44 | 84.2 [80.3–87.7] | 14 (1.2) |
| Acute otitis media | 60 | 60 (100) | 6 (5–7) | 29 | 48.3 [35.2–61.6] | 39 (3.3) |
| Congestive otitis or serous otitis | 10 | 5 (50) | 3 (0–7) | 5 | 50.0 [18.7–81.3] | 36 (3.0) |
| Acute sinusitis | 41 | 39 (95.1) | 5 (5–7) | 20 | 48.8 [32.9–64.9] | 26 (2.2) |
| **Total** | **714** | **334 (46.8)** | **0 (0–6)** | **476** | **66.7 [63.1–70.0]** | **1194 (100)** |

COPD = chronic obstructive pulmonary disease. RADT = rapid antigen detection test, ATB = antibiotic, IQR = interquartile range

*Recommended durations used for evaluation: see Table 1

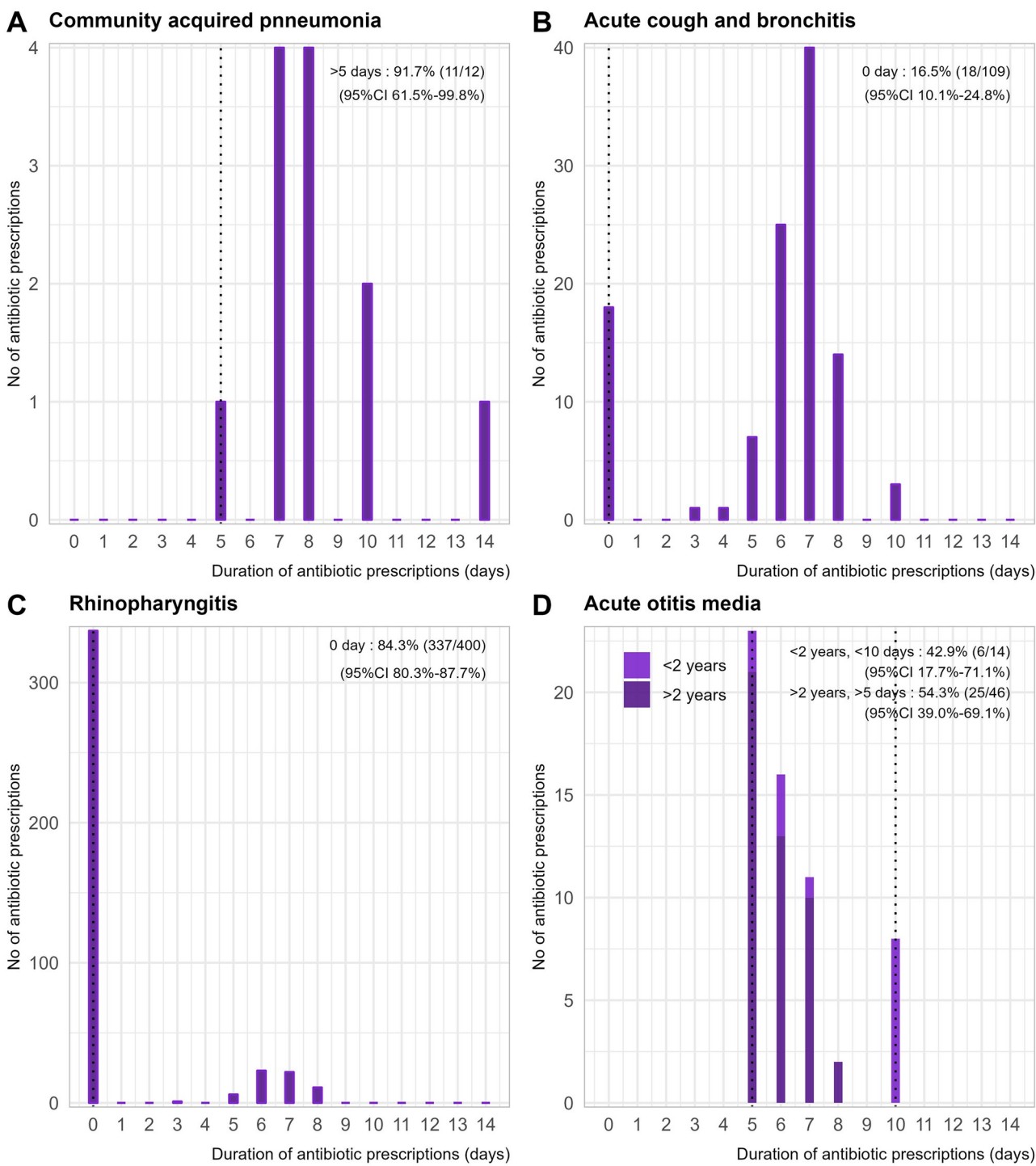

**Fig 1. Duration of antibiotic prescriptions for: A-Community acquired pneumonia (N = 12), B-Acute cough and bronchitis (N = 109), C-Rhinopharyngitis (N = 400), D-Acute otitis media (N = 70).** Vertical dots indicate durations recommended in France [12].

RADT was not performed. In cases of acute otitis media in children under two years of age, the duration was shorter than the recommended 10 days in 6 out of 14 cases (42.9%) (Fig 1D).

In total, the study identified 1,194 days of excess antibiotic treatment, averaging 1.7 days per prescription [95% CI: 1.4–1.9]. The majority of excess days were associated with acute cough and bronchitis (609 days, accounting for 51.0% of total excess days) and nasopharyngitis (413 days, accounting for 34.6% of the total).

## Discussion

In this cross-sectional study, we evaluated 714 consultations for upper and lower respiratory infections in children and adults. Nearly half of these consultations resulted in antibiotic prescriptions, with durations conforming to current French recommendations in two-thirds of cases. However, this overall result masks significant disparities depending on the indication. For nasopharyngitis, 80% of prescribed durations aligned with recommended guidelines, but due to the high frequency of this infection, it accounted for 35% of excess antibiotic prescription durations. In cases of acute bronchitis and cough, antibiotics were prescribed in 83.5% of consultations, with a median duration of 7 days, making this indication responsible for about half of all excess prescriptions. Overall, we identified 1,194 excess days of antibiotic use, averaging 1.7 extra days per consultation. When extrapolated to the approximately 100,000 general practitioners in France, our findings suggest that in March 2022 alone, roughly 10 million excess antibiotic days may have been prescribed. These results underscore significant opportunities for improvement in reducing both the initiation and duration of antibiotic treatments.

The evaluation of antibiotic treatment duration in primary care has been relatively underexplored. In a recent study conducted in the United States [12], the median duration of antibiotic therapy was 10 days for all indications, except for acute cystitis, which had a median duration of 7 days. Overall, in 55% of cases, the prescribed duration exceeded the minimum recommended duration for effective treatment. The same authors found that more than twothirds of antibiotic treatments for acute sinusitis in adults lasted 10 days or more, despite national recommendations of 5 to 7 days for uncomplicated cases [13]. In a Canadian study examining nasopharyngitis, skin or soft tissue infections, and pneumonia, 37.8% of antibiotics prescribed for bacterial infections were potentially inappropriate, with 19.6% having durations longer than recommended [14]. In a large English cross-sectional study [15], nearly one million antibiotic treatments prescribed for various indications (respiratory, urinary, and skin infections) were evaluated. The most common indications for antibiotic treatment were acute coughs and bronchitis, accounting for 41.6% of the consultations studied. Over 80% of these treatment durations exceeded national recommendations. The average excess duration per visit was 1.4 days, though this figure did not consider cases where antibiotics may have been unnecessarily prescribed. These results were confirmed even when the study was restricted to patients without comorbidities and with no history of immunosuppressive or corticosteroid therapy, indicating that these factors did not influence the decision to prescribe treatment beyond the recommended duration [15].

Our study appears to confirm that one of the main challenges in reducing antibiotic use in primary care lies in the management of acute cough and bronchitis. General practitioners often face patient pressure and find it challenging to refrain from prescribing antibiotics. Acute bronchitis is a lower respiratory tract infection characterized primarily by cough, with or without sputum, that typically resolves spontaneously within three weeks, without recent clinical or radiographic findings indicating an alternative diagnosis. The recent French recommendations [11] appear challenging to apply universally in primary care, particularly as they do not address specific cases such as elderly or frail patients. This difficulty arises, in part, due

to the limited clinical and paraclinical elements available for general practitioners to distinguish between viral bronchitis and acute pneumonia [16]. The presence or absence of stained sputum is not a reliable indicator for differentiating between bacterial and viral lower respiratory tract infections. Only the detection of an infiltrate on chest radiography, which is often not immediately accessible, can confirm a definitive diagnosis of pneumonia.

The latest English recommendations appear to be more adapted to real-world practice [17]. They acknowledge the current clinical limitations and the resulting uncertainty in determining the etiology, while providing a clearer framework for first-line antibiotic prescriptions by listing comorbidities that help identify patients at risk of complications, such as progression to acute pneumonia, thereby justifying antibiotic use. Additionally, they recommend implementing a deferred prescription strategy or utilizing C-reactive protein measurement when needed. In all cases, the duration of antibiotic therapy should not exceed five days.

In the United States, the American College of Physicians (ACP) also advises against initiating antibiotic treatment for patients with acute bronchitis, while emphasizing the importance of evaluating their immune status. Routine tests or medications are not currently recommended for immunocompetent adult outpatients presenting with this condition for the first time, due to a lack of evidence supporting their effectiveness in reducing the severity or duration of the cough. If the cough associated with acute bronchitis persists or worsens, the patient should be reassessed, at which point the physician may consider targeted investigations and prescribe an antibiotic if a bacterial infection is suspected [18].

The results for nasopharyngitis are more encouraging; however, the frequency of this condition as a reason for consultation can lead to potentially high volumes of antibiotic prescriptions on a collective scale. For sore throats, it is notable that when the RADT was negative, no antibiotics were prescribed, and when the RADT was positive, treatment durations generally adhered to guidelines. However, the RADT was used in only 30% of sore throat cases in the study, indicating that efforts to promote the use of this diagnostic tool should be reactivated.

Our study has several limitations. It was conducted in only two primary care centers, which raises questions about its external validity. Additionally, since the study was carried out in routine care settings, we did not seek confirmation of the diagnoses made by the general practitioners through complementary examinations. We also included only the first consultations for upper and lower respiratory infections, which may have led to an underestimation of prescription durations; it is possible that GPs may be more inclined to prescribe antibiotics during follow-up visits for persistent infections. Nonetheless, one of the study's strengths is the comprehensiveness and reliability of the data collected: during the study period, all medical records of the participating GPs were reviewed.

The fight against antibiotic resistance requires reducing unnecessary antibiotic use, and our study demonstrates that there is significant room for improvement in France. Shortening antibiotic therapy for common infections in primary care not only helps to decrease selection pressure, which is expected to have beneficial effects, but also reduces costs and side effects, while improving therapeutic compliance [11].

General practitioners, often facing uncertainty about the progression of initially viral conditions in certain patients and pressure from patients to prescribe antibiotics, need tools to help them adhere more closely to current recommendations. Reducing inappropriate antibiotic prescribing will require multifaceted actions involving healthcare professionals, including patient education, prescription audits with feedback, financial incentives for adherence to best practices, access to diagnostic tools, unit dispensing of antibiotics by pharmacies to ensure adherence to prescribed durations, and promoting the new paradigm of reducing antibiotic durations—"shorter is better" [19]. This approach should reassure healthcare professionals

that shorter durations do not increase the risk of complications, while maintaining cure rates and disease resolution times.

## Supporting information

**S1 Data. Minimal data set.**
(CSV)

## Acknowledgments

The authors would like to thank all the GPs who took part in the study.

## Author Contributions

**Conceptualization:** Emmanuel Piednoir, Marianne Delestre, Pascal Thibon.

**Data curation:** Simon Ferme.

**Formal analysis:** Pascal Thibon.

**Investigation:** Simon Ferme.

**Methodology:** Simon Ferme, Emmanuel Piednoir, Renaud Verdon, Pascal Thibon.

**Writing – original draft:** Simon Ferme, Pascal Thibon.

**Writing – review & editing:** Emmanuel Piednoir, Marianne Delestre, Elise Fiaux, Johann Masik, Renaud Verdon, Pascal Thibon.

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
