## [Decision Letter · Decision Letter 0]

25 Sep 2024

PONE-D-24-07363Evaluation of antibiotic treatment initiation and duration practices in primary care. Cross-sectional study in two French multi-professional health centers.PLOS ONE

Dear Dr. Thibon,

Thank you for submitting your manuscript to PLOS ONE. After careful consideration, we feel that it has merit but does not fully meet PLOS ONE’s publication criteria as it currently stands. Therefore, we invite you to submit a revised version of the manuscript that addresses the points raised during the review process.

Please revise and correct the grammar of the manuscript. The authors may wish to consult a native English speaker to review the manuscript. Please provide more recent data in the introduction on deaths attributed to infections/ antibiotic resistance.

We look forward to receiving your revised manuscript.

Kind regards,

Petra Czarniak, PhD

Academic Editor

PLOS ONE

 Journal Requirements: When submitting your revision, we need you to address these additional requirements. 1. Please ensure that your manuscript meets PLOS ONE's style requirements, including those for file naming. The PLOS ONE style templates can be found at https://journals.plos.org/plosone/s/file?id=wjVg/PLOSOne_formatting_sample_main_body.pdf and https://journals.plos.org/plosone/s/file?id=ba62/PLOSOne_formatting_sample_title_authors_affiliations.pdf 2. Thank you for stating the following in your Competing Interests section: NO Please complete your Competing Interests on the online submission form to state any Competing Interests. If you have no competing interests, please state ""The authors have declared that no competing interests exist."", as detailed online in our guide for authors at http://journals.plos.org/plosone/s/submit-now   This information should be included in your cover letter; we will change the online submission form on your behalf. 3. In the online submission form, you indicated that "Data will be provided upon request to the authors" All PLOS journals now require all data underlying the findings described in their manuscript to be freely available to other researchers, either 1. In a public repository, 2. Within the manuscript itself, or 3. Uploaded as supplementary information.This policy applies to all data except where public deposition would breach compliance with the protocol approved by your research ethics board. If your data cannot be made publicly available for ethical or legal reasons (e.g., public availability would compromise patient privacy), please explain your reasons on resubmission and your exemption request will be escalated for approval.

Reviewers' comments:

Reviewer's Responses to Questions

**Comments to the Author**

1. Is the manuscript technically sound, and do the data support the conclusions?

Reviewer #1: Partly

Reviewer #2: Yes

2. Has the statistical analysis been performed appropriately and rigorously? 

Reviewer #1: I Don't Know

Reviewer #2: Yes

3. Have the authors made all data underlying the findings in their manuscript fully available?

Reviewer #1: Yes

Reviewer #2: Yes

4. Is the manuscript presented in an intelligible fashion and written in standard English?

Reviewer #1: No

Reviewer #2: Yes

5. Review Comments to the Author

Reviewer #1: This study attempts to identify and rectify the misuse of antibiotics for the management/treatment of upper and lower respiratory tract infections. I think studies like this are critical while tackling the ever-evolving issue of antibiotic resistance. Though the methodology and data presentation seem to be scientifically sound, I have major issues with the written English. I couldn’t follow through some of the paragraphs. There are major grammatical and syntax issues throughout the manuscript, making it very confusing to the reader. Some sentences are way too long, thereby making them incomprehensible. From as much as I was able to comprehend, I think this work is worth consideration. However, I highly recommend the authors to consult a native English speaker to review this work before it can actually be considered for publication.

Suggestion: I think it would make this work better if the authors included a discussion on the most misused antibiotics (with names) from their findings.

Reviewer #2: Thank you for the opportunity to review this article. This is an interesting paper and an important area evaluating the appropriateness of antibiotic prescribing across two French multi-professional health centers. The paper is well written, and I have provide some suggestions that the authors might like to consider.

Introduction:

• Are there more recent data than 2015/2016?

Methods:

• How did you extract the data from the practice record? Or did you ask GPs to manually record each consultation?

• Table 1 - what is the guideline recommendations for CAP and OM?

• How many participants refused to participate? How did you record the total number of consultations including those who refused to participate? It is unclear how you obtained the data, which fields did you use to make your assessment? Did you obtain any identifiable information? Did you have access to progress notes that might provide more information for the assessment?

Results:

• Did you include repeats in the assessment? This may be included as part of duration?

• Is it possible to split the adults and children data in the analysis? It may be interesting to see whether children are prescribed antibiotics more or less often than adults for these conditions.

Discussion:

• Often guidelines are only recommendations, what about reasons why GPs are not concordant? It might be good to find out more i.e. patient pressure, diagnostic uncertainty, public holidays and long weekends leading to antibiotic prescribing rather than waiting etc.

• How long did it take to complete your assessment? have you considered providing timely feedback reporting?

• Did you consider delayed prescribing and other factors in calculating duration? In Australia, we have issues with pack size, lack of recorded instructions to patients, and delayed prescribing (even though they may not be filled, but still considered as prescribed) leading to excessive duration

• Were there any antibiotics incorrectly prescribed? Even if antibiotics prescribed were not recommended, could they be a considered as reasonable acceptable choice?

• Did you take account any allergy assessment?

• Can you provide any reasons for GPs not performing RADT? due to cost?

Minor points

• Please spell out acronyms in the first instance, i.e. WHO, FDA etc. Is it correct that is it the American FDA or is it US FDA?

• This paper may be of interest https://www1.racgp.org.au/ajgp/2022/january-february/antimicrobials-for-respiratory-infections-1/

Hope my comments helped. Good luck with your paper!

6. PLOS authors have the option to publish the peer review history of their article (what does this mean?). If published, this will include your full peer review and any attached files.

Reviewer #1: **Yes: **John Jacob

Reviewer #2: **Yes: **Ruby Biezen

---

## [Author Response · Author response to Decision Letter 0]

15 Nov 2024

PONE-D-24-07363

Evaluation of antibiotic treatment initiation and duration practices in primary care. Cross-sectional study in two French multi-professional health centers.

Academic Editor’s comments

> Thank you very much for reviewing our work. Below, please find our responses to your comments as well as those of the reviewers.

Please revise and correct the grammar of the manuscript. The authors may wish to consult a native English speaker to review the manuscript.

> We have thoroughly revised the manuscript, incorporating your comments and suggestions.

Please provide more recent data in the introduction on deaths attributed to infections/ antibiotic resistance.

>Done. We have included AMR-attributable mortality data for 2021 from the Lancet (2024) and have removed the outdated 2015 French data.

Reviewer #1 (Comments for the Author):

This study attempts to identify and rectify the misuse of antibiotics for the management/treatment of upper and lower respiratory tract infections. I think studies like this are critical while tackling the ever-evolving issue of antibiotic resistance.

>Thank you very much for your comment.

Though the methodology and data presentation seem to be scientifically sound, I have major issues with the written English. I couldn’t follow through some of the paragraphs. There are major grammatical and syntax issues throughout the manuscript, making it very confusing to the reader. Some sentences are way too long, thereby making them incomprehensible. From as much as I was able to comprehend, I think this work is worth consideration. However, I highly recommend the authors to consult a native English speaker to review this work before it can actually be considered for publication.

> We apologize for any inconvenience. We have thoroughly revised the manuscript, taking your comments into full consideration.

Suggestion: I think it would make this work better if the authors included a discussion on the most misused antibiotics (with names) from their findings.

> Unfortunately, we are unable to produce these results, as the study specifically focused on treatment durations rather than the appropriateness of the prescribed molecules.

Reviewer #2 (Comments for the Author):

Thank you for the opportunity to review this article. This is an interesting paper and an important area evaluating the appropriateness of antibiotic prescribing across two French multi-professional health centers. The paper is well written, and I have provide some suggestions that the authors might like to consider.

>Thank you very much for your comment, and for your suggestions.

Introduction:

• Are there more recent data than 2015/2016?

> We have included AMR-attributable mortality data for 2021 from the Lancet (2024) and have removed the outdated 2015 French data.

Methods:

• How did you extract the data from the practice record? Or did you ask GPs to manually record each consultation?

> At the end of the study period, a query was conducted using the GPs' software to extract data on patient characteristics, indications, and antibiotics prescribed. We have added this clarification to the manuscript.

• Table 1 - what is the guideline recommendations for CAP and OM?

> For community-acquired pneumonia (CAP), the guideline specifies: if clinical improvement is observed upon reassessment at day 3 (D+3) — indicated by apyrexia and improved vital signs — treatment should be continued for 5 days. If no improvement is noted at D+3, the maximum duration is 7 days (for CAP cases hospitalized in intensive care units: 7 days, or 5 days if there is clinical improvement).

For otitis media (OM): 5 days of treatment is recommended, with a duration of 10 days for children under 2 years of age.

• How many participants refused to participate? How did you record the total number of consultations including those who refused to participate? It is unclear how you obtained the data, which fields did you use to make your assessment? Did you obtain any identifiable information? Did you have access to progress notes that might provide more information for the assessment?

> The study was exhaustive, as it relied on data extraction from each doctor's software, encompassing patient characteristics, type of infection, and treatment. No patient refused to participate, and we have added this clarification to the manuscript.

Results:

• Did you include repeats in the assessment? This may be included as part of duration?

> If a patient had multiple visits for the same infection, only the first consultation was included, ensuring there were no repeated entries.

• Is it possible to split the adults and children data in the analysis? It may be interesting to see whether children are prescribed antibiotics more or less often than adults for these conditions.

> Thank you for the interesting suggestion. We separated the analysis for otitis media (OM) since the duration recommendations differ for children under 2 years old. However, a child/adult subgroup analysis was not part of the study protocol for other types of infections. We do plan to conduct further analyses along these lines, but we are unable to include them in the current manuscript.

Discussion:

• Often guidelines are only recommendations, what about reasons why GPs are not concordant? It might be good to find out more i.e. patient pressure, diagnostic uncertainty, public holidays and long weekends leading to antibiotic prescribing rather than waiting etc.

> Indeed, there is much to consider regarding the reasons for deviations from the recommendations. Our study was purely quantitative and we did not explore these reasons with the participating GPs. However, we did touch upon some of these aspects in the manuscript. For instance, we discussed factors such as patient pressure and diagnostic uncertainty, particularly in elderly or frail patients presenting with cough. We also addressed challenges related to communicating the new French recommendations to prescribers.

• How long did it take to complete your assessment? have you considered providing timely feedback reporting?

>The study lasted 6 weeks in each practice, and feedback was provided at its conclusion.

• Did you consider delayed prescribing and other factors in calculating duration? In Australia, we have issues with pack size, lack of recorded instructions to patients, and delayed prescribing (even though they may not be filled, but still considered as prescribed) leading to excessive duration

> Delayed prescriptions are still infrequently used in France, and none were observed in our study.

• Were there any antibiotics incorrectly prescribed? Even if antibiotics prescribed were not recommended, could they be a considered as reasonable acceptable choice?

> The study focused solely on treatment durations and did not assess the specific molecules prescribed. It is important to note that French recommendations on antibiotic duration do not specify particular molecules, except in special cases, such as Group A streptococcus pharyngitis in patients with a penicillin allergy and sinusitis.

• Did you take account any allergy assessment?

> Allergy history was collected to assess treatment durations, particularly for the specific case of Group A streptococcus pharyngitis.

• Can you provide any reasons for GPs not performing RADT? due to cost?

> In France, rapid antigen tests (RATDs) are provided free of charge to general practitioners by the national health insurance system. The primary reasons for their non-use are associated with the increased duration of consultations.

Minor points

• Please spell out acronyms in the first instance, i.e. WHO, FDA etc. Is it correct that is it the American FDA or is it US FDA?

>Done. It is the US FDA, not the American FDA (thank you!).

• This paper may be of interest https://www1.racgp.org.au/ajgp/2022/january-february/antimicrobials-for-respiratory-infections-1/

> We found it very interesting to read this study based on a substantial sample size. We are also involved in the development of a data warehouse in France, which we hope will allow us to gather data from large populations, including treatment indications, similar to your approach.

---

## [Editor Report · Decision Letter 1]

21 Nov 2024

Evaluation of antibiotic treatment initiation and duration practices in primary care. Cross-sectional study in two French multi-professional health centers.

PONE-D-24-07363R1

Dear Dr. Thibon,

We’re pleased to inform you that your manuscript has been judged scientifically suitable for publication and will be formally accepted for publication once it meets all outstanding technical requirements.

Kind regards,

Petra Czarniak, PhD

Academic Editor

PLOS ONE
---

## [Editor Report · Acceptance letter]

29 Nov 2024

PONE-D-24-07363R1 

PLOS ONE

Dear Dr. Thibon, 

I'm pleased to inform you that your manuscript has been deemed suitable for publication in PLOS ONE. Congratulations! Your manuscript is now being handed over to our production team.

Kind regards, 

on behalf of

Dr. Petra Czarniak 

Academic Editor

PLOS ONE